# Human In Vitro Oxidized Low-Density Lipoprotein (oxLDL) Increases Urinary Albumin Excretion in Rats

**DOI:** 10.3390/ijms25105498

**Published:** 2024-05-17

**Authors:** Kamil Dąbkowski, Ewelina Kreft, Kornelia Sałaga-Zaleska, Gabriela Chyła-Danił, Agnieszka Mickiewicz, Marcin Gruchała, Agnieszka Kuchta, Maciej Jankowski

**Affiliations:** 1Department of Clinical Chemistry, Medical University of Gdańsk, 80-210 Gdańsk, Poland; kamil.dabkowski@gumed.edu.pl (K.D.); ewelina.kreft@gumed.edu.pl (E.K.); kornelia.salaga-zaleska@gumed.edu.pl (K.S.-Z.); gabriela_chyla@gumed.edu.pl (G.C.-D.); agakuchta@gumed.edu.pl (A.K.); 21st Department of Cardiology, Medical University of Gdańsk, 80-210 Gdańsk, Poland; agnieszka.mickiewicz@gumed.edu.pl (A.M.); marcin.gruchala@gumed.edu.pl (M.G.)

**Keywords:** albuminuria, kidney, oxidized low-density lipoprotein, nephrin, oxidative stress

## Abstract

Hypercholesterolemia-associated oxidative stress increases the formation of oxidized low-density lipoprotein (oxLDL), which can affect endothelial cell function and potentially contribute to renal dysfunction, as reflected by changes in urinary protein excretion. This study aimed to investigate the impact of exogenous oxLDL on urinary excretion of albumin and nephrin. LDL was isolated from a patient with familial hypercholesterolemia (FH) undergoing lipoprotein apheresis (LA) and was oxidized in vitro with Cu (II) ions. Biochemical markers of LDL oxidation, such as TBARS, conjugated dienes, and free ε-amino groups, were measured. Wistar rats were treated with a single intraperitoneal injection of PBS, LDL, or oxLDL (4 mg of protein/kg b.w.). Urine was collected one day before and two days after the injection. We measured blood lipid profiles, urinary protein excretion (specifically albumin and nephrin), and markers of systemic oxidative stress (8-OHdG and 8-iso-PGF2α). The results showed that injection of oxLDL increased urinary albumin excretion by approximately 28% (310 ± 27 μg/24 h vs. 396 ± 26 μg/24 h, *p* = 0.0003) but had no effect on nephrin excretion. Neither PBS nor LDL had any effect on urinary albumin or nephrin excretion. Additionally, oxLDL did not affect systemic oxidative stress. In conclusion, hypercholesterolemia may adversely affect renal function through oxidatively modified LDL, which interferes with the renal handling of albumin and leads to the development of albuminuria.

## 1. Introduction

Hypercholesterolemia is a condition characterized by elevated levels of low-density lipoprotein (LDL) and often associated with the early stages of atherosclerosis, which is characterized by excessive cholesterol accumulation, oxidative stress, endothelial dysfunction, and inflammation that can impair vascular function, including renal function [1]. Data from a cohort study of patients with familial hypercholesterolemia (FH) suggest that excess cholesterol accumulation in the kidneys may contribute to the development of chronic kidney disease [2]. LDL is a major carrier of plasma cholesterol and has a highly heterogeneous nature. The lipid core of this structure is composed of triglycerides and cholesteryl esters, while the surface monolayer is made up of phospholipids and a single copy of apolipoprotein B100. Apolipoprotein B100 serves as a ligand for hepatic LDL receptors (LDLRs) [3]. Mutations in the LDLR gene are responsible for causing FH, which is the most severe form of genetic hypercholesterolemia [4]. In patients with hypercholesterolemia, LDL particles undergo oxidative modification, resulting in the formation of small lipoproteins and an increase in the synthesis of oxidized LDL (oxLDL) to a range of 1–8% of the total LDL volume [5]. OxLDL is a lipoprotein fraction that is highly reactive and has been linked to endothelial cell dysfunction [6]. It is generally accepted that oxidative modification of LDL results in loss of recognition by the LDL receptor and a shift to recognition by scavenger receptors, which do not have reduced surface expression when exposed to excess cholesterol [7]. It has been observed that scavenger receptors like CXCL16 are expressed on podocytes and may play a role in their apoptosis [8,9]. There is considerable evidence that oxLDL is a critical factor in the development of atherosclerosis and renal disease. Studies have demonstrated that oxLDL has a higher affinity for rat glomeruli in vivo and cultured mesangial cells in vitro [10]. OxLDL was detected in renal biopsies from human subjects [11]. The presence of OxLDL in the glomeruli of rats with experimental focal segmental glomerulosclerosis is considered a hallmark of such a progressive glomerular injury that leads to disturbances in the permeability of the glomerular filtration barrier to protein [12].

The filtration barrier permits the passage of water and small solutes into Bowman’s space. It is composed of a basement membrane and a monolayer of endothelial cells and podocyte foot processes. Adjacent foot processes are bridged by a slit diaphragm with nephrin as a key component. The slit diaphragm is a crucial layer that limits the passage of solutes based on their molecular size. Its loss is believed to be a significant factor in the advancement of kidney disease [13]. A systematic review and meta-analysis suggest that nephrin excretion in urine may be a useful tool for detecting early glomerular injury [14]. Small amounts of albumin are filtered across the glomerulus into Bowman’s space. Subsequently, this albumin is reabsorbed in the proximal tubule after binding to the apical megalin–cubilin receptor. It can then be transcytosed, degraded by lysosomal enzymes, or exocytosed into the tubular fluid [15].

It is possible that a disruption in podocyte number or function may lead to an increase in protein, such as albumin, passing through the glomerular filter into the urine, which could affect tubular cell function and result in an increased concentration of albumin in the urine. It is important to note that albumin exerts its toxic effects when it is degraded and reabsorbed by proximal tubular epithelial cells [16]. Albuminuria, which is defined as urinary albumin excretion exceeding 30 mg of albumin/g of creatinine, has been found to be associated with hypercholesterolemia [17]. It is important to note that increased urinary albumin excretion is a marker of microvascular disease and associated with increased cardiovascular morbidity and mortality, even at levels not considered pathological, i.e., 10–30 mg of albumin/g of creatinine [18]. Furthermore, the hypothesis that cholesterol causes renal toxicity is supported by the results of cholesterol-lowering therapy using statins [19]. Additionally, the physical removal of LDL from the blood through LA has been shown to ameliorate albuminuria and prevent podocyte loss in renal patients [20]. Thus, oxLDL, which is produced in increased amounts in hypercholesterolemia, may directly interact with the kidneys, leading to changes in the excretion of certain proteins in the urine. This study investigated the impact of oxLDL on urinary albumin and nephrin excretion in the absence of systemic oxidative stress. The results obtained suggest that oxLDL increases the urinary excretion of albumin while having no effect on the excretion of nephrin.

## 2. Results

The clinical and biochemical characteristics of the analyzed patients are shown in Table 1.

Table 2 shows the composition of LDL isolated from patients with FH who underwent LA. The lipid/protein ratio of the isolated LDL is 3.5 ± 0.1, and the ratio of esterified cholesterol to triacylglycerols is 5.9 ± 0.2. Parameters indicating the efficacy of in vitro LDL oxidation and the formation of oxidized LDL (oxLDL) are shown in Table 3. These samples show higher levels of TBARS and conjugated dienes compared to native LDL (nLDL), with 7-fold and 3.2-fold increases, respectively. Additionally, the levels of free ε-amino groups are 2.7 times lower than those found in nLDL.

Injection of PBS and LDL did not significantly affect urinary albumin excretion (U_alb_), yielding the following results: 408 ± 52 μg/24 h vs. 387 ± 32 μg/24 h, *p* = 0.554, as shown in Figure 1A, and 371 ± 27 μg/24 h vs. 321 ± 27 μg/24 h, *p* = 0.574, as shown in Figure 1B, respectively. However, the injection of oxLDL resulted in a significant increase in U_alb_ (310 ± 27 μg/24 h vs. 396 ± 26 μg/24 h, *p* = 0.0003, as shown in Figure 1C). Urinary nephrin excretion (U_neph_) was not affected by injections of PBS (2.53 ± 0.74 ng/24 h vs. 2.07 ± 0.67 ng/24 h, *p* = 0.639, Figure 1D), LDL (1.24 ± 0.14 ng/24 h vs. 1.75 ± 0.61 ng/24 h, *p* = 0.489, Figure 1E), or oxLDL (0.50 ± 0.11 ng/24 h vs. 0.99 ± 0.21 ng/24 h, *p* = 0.173, Figure 1F). Injection of PBS, LDL, or oxLDL had no significant effect on serum concentrations of cholesterol, triglycerides, TBARS, or creatinine clearance, as shown in Table 4. In addition, injection of LDL or oxLDL did not affect urinary excretion of water, TBARS, 8-iso-PGF2α, or 8-OHdG, as shown in Table 5.

The tested rats received intraperitoneal injections of PBS and either LDL or oxLDL at a dose of 4 mg of LDL protein per kg of body weight. The results display single data points of urinary albumin (Figure 1A–C and nephrin(Figure 1D–F)) excretion obtained before (preinjection) and two days after injection (postinjection). Statistical significance was determined using a paired-samples *t*-test (* *p* as indicated).

## 3. Discussion

One question raised by this study is whether oxidized LDL can increase urinary excretion of albumin, which is known to have nephrotoxic properties and is also a predictor of both cardiovascular and non-vascular mortality in the general population. The main finding of this study suggests that a single intraperitoneal injection of in vitro oxidized human LDL (oxLDL) increases urinary albumin excretion by approximately 28% while having no effect on urinary nephrin excretion. In addition, injection of oxLDL does not appear to affect lipid profiles or systemic stress levels.

In this study, low-density lipoprotein was isolated from patients with FH who underwent LA using the MONET system. This LA system includes a filter that selectively allows molecules with a molecular weight of less than 100 kDa to pass through while retaining lipoproteins by reducing permeability for molecules with a molecular weight of more than 1000 kDa. After the MONET filter washing procedure, the waste fluid was enriched with LDL and subsequently utilized for further LDL isolation. FH, caused by a genetic defect in the uptake of LDL by the LDL receptor, may result in abnormal LDL composition through hypocatabolism, which increases the half-life of LDL and the mean age of circulating particles in plasma [4]. LDL in elevated concentrations can undergo oxidative modification in vivo, in part due to increased endothelial superoxide anion production [21,22]. It is important to note that older plasma lipoproteins are more susceptible to oxidation [23]. It has been observed that LDL obtained from patients with FH shows an increase in cholesterol ester and free cholesterol levels [24,25]. Furthermore, it has been noted that LDL obtained from FH patients has an increased ratio of cholesterol to phospholipid and a decreased ratio of lecithin to sphingomyelin [26].

In this study, it was found that the composition of LDL isolated from aphaeresis fluid after filter washing consisted of 22% proteins and 78% lipids. This study found that the oxidation of isolated LDL was effective. It was observed that the in vitro oxidation of isolated LDL using Cu (II) ions had an impact on chemical parameters such as thiobarbituric-acid-reactive substances (TBARS), conjugated dienes, and free ε-amino groups.

Due to the relatively low permeability of the glomerular filtration barrier to albumin and the efficient reabsorption mechanisms in the proximal tubule, urinary albumin excretion is generally kept at a low level [15,27]. According to our findings, it appears that the rise in urinary albumin excretion caused by oxLDL may be attributed to an increase in albumin leakage through the glomerular filtration barrier and/or a decrease in albumin reabsorption from tubule fluid in the proximal tubule. It is important to note that the fractional clearance of LDL in normal rat kidneys is extremely low, which is consistent with a normally functioning size-selective glomerular basement. As a result, the concentrations of oxLDL present in urine are very low [28]. Therefore, the interference of injected oxLDL with the reabsorption process of albumin in the proximal tubule may be negligible. It appears that albumin has a greater effect on glomerular permeability to albumin than on albumin reabsorption in the proximal tubule. It is worth noting that there was no observed increase in nephrin excretion following oxLDL injection, which suggests that podocyte injury may not be the primary cause. However, it is possible that scavenger receptors on podocytes could interact with oxLDL, resulting in alterations to glomerular permeability to albumin [8]. Taken together, it appears that oxLDL may impact urinary albumin excretion by affecting the endothelial cell layer of the glomerular filter. Endothelial cells express scavenger receptors that facilitate the uptake and internalization of oxLDL as membrane-bound carriers [29]. Oxidized LDL has been found to lead to endothelial activation, dysfunction, and injury [6]. Activation of endothelial cells can lead to the expression of various genes, such as endothelin, tissue factor, cyclooxygenase, and nitric oxide synthase [30]. It is possible that the above factors may affect glomerular permeability to albumin. The available evidence suggests that the effect of oxLDL is not caused by the induction of systemic stress. This is supported by the observation that urinary concentrations of oxidative stress markers—TBARS, 8-iso-PGF2α, and 8-OHdG [31,32]—do not appear to be significantly affected by oxLDL injections (Table 4).

We recognize the potential limitations of this study, including the following: (1) intraperitoneal injection may not accurately mimic the continuous oxLDL concentration in plasma passing through the glomerular filter; (2) only a single dose of LDL was studied, and future studies should define an optimal, pathophysiologically relevant dose range; (3) the variable composition of LDL requires optimization by using different sources of LDL; and (4) the mechanisms underlying the potentiating effects of oxLDL on the glomerular filter require investigation.

## 4. Materials and Methods

### 4.1. LDL Isolation

Low-density lipoproteins (LDL) were isolated from apheresis fluid obtained during LA with the MONET filter in patients with FH at the 1st Chair and Department of Cardiology, Medical University of Gdansk. Sequential density gradient ultracentrifugation was performed using a Optima TLX-120 ultracentrifuge equipped with a TLA-100.3 fixed-angle rotor (Beckman Coulter Inc., Boulevard Brea, IN, USA) using slow acceleration and deceleration [33]. Apheresis fluid obtained by washing the MONET filters (Guyancourt, France) was transferred to 3.5 mL polycarbonate centrifuge tubes, and a discontinuous density gradient was established by adding to the apheresis fluid (2.1 mL) a solution with a density of 1.006 g/mL and the following composition: 195 mM of NaCl, 10 mg/dL of EDTA, and 1 mM of NaOH. The tubes were ultracentrifuged at 541,000× *g* for 1 h at 4 °C. The upper fraction was removed, and the lower fraction was mixed with 16.7% NaCl in a 1:1 ratio. The appropriate density fraction was then centrifuged at 541,000× *g* for 2.5 h at 16 °C, after which the LDL-rich fraction was collected from the top of the tube (d < 1.063 g/mL). The LDL-rich fraction was desalted by passing the samples through a Sephadex G-10 column (GE Healthcare, Chalfont St. Giles, Buckinghamshire, UK) and filtered through a 0.20 μm pore-size syringe filter (Corning Inc., New York, NY, USA. Finally, sucrose was added to a final concentration of 10%, and the samples were stored at −80 °C.

### 4.2. Copper-Dependent Oxidation of LDL

LDL (1 mg/mL) was oxidized in PBS containing 5 µM of CuSO_4_ at 37 °C for 24 h, following a method that has been previously described [34]. To stop the oxidation reaction, 0.5 mM of EDTA (pH 8.5) and 50 μM of butylated hydroxytoluene were added. The samples were dialyzed at 4 °C against PBS, with three buffer changes of 8 h each. At the time of the first buffer change, 0.5 mM of EDTA was added. Levels of thiobarbituric-acid reactive substances (TBARS), conjugated dienes, and free ε-amino groups were measured to assess LDL oxidation. The LDL fractions (50 μL) were resuspended in 400 μL of 150 mM NaCl. Subsequently, the suspension was combined with 150 μL of 20% trichloroacetic acid and 150 μL of 0.67% thiobarbituric acid/glacial acetic acid (1:1, *v*/*v*). The mixture was heated to 95 °C for 60 min, cooled on ice, and then shaken with 600 µL of n-butanol:pyridine (15:1, *v*/*v*) for 20 s. After centrifugation at 4000× *g* for 10 min at 4 °C, the fluorescence of the extracted solution was measured using excitation and emission wavelengths of 515 and 550 nm, respectively, on a black 96-well microplate (Cytation™ 3, BioTek Instruments Inc., Winooski, VT, USA). Measurement of peroxide formation was performed using the conjugated diene assay based on absorbance at 234 nm with a final concentration of 100 µg of protein/mL. The assessment of the free ε-amino groups in the LDL protein was carried out through the utilization of trinitrobenzenesulfonic acid (TNBS). The LDL samples were incubated with 0.01% TNBS in a reactive buffer (0.1 M Na_2_CO_3_, pH 8.5) for a period of 2 h at 37 °C. The reaction was stopped by adding a final concentration of 2% SDS, 0.1 HCl, and the absorbance at 335 nm was measured. The concentration of ε-amino groups was determined by referring to an L-valine standard [35].

### 4.3. Animals

Wistar rats were obtained from the Tri-City Academic Centre for Laboratory Animals’ Research and Service Centre (Medical University of Gdańsk, Poland).

### 4.4. Experimental Protocol

Male Wistar rats weighing 200–250 g and aged 8–10 weeks were used in the studies. The rats were maintained on a 12 h light/12 h dark cycle and fed a standard pellet diet (Labofeed B, Zofia Połczyńska Wytwórnia Pasz-Morawski, Poland) with ad libitum access to water. The rats were divided into the following three groups (n = 4 in each group), each receiving intraperitoneal injections:PBS group, receiving phosphate-buffered saline injections;LDL group, receiving LDL injections (4 mg of LDL protein/kg of b.w.);oxLDL group, receiving oxidized LDL injections (4 mg of LDL protein/kg of b.w.).

The doses of LDL and oxLDL were administered in accordance with previously published data [36]. The endotoxin content of the lipoprotein preparations was tested using the ToxinSensor™ Chromogenic LAL Endotoxin Assay Kit (GenScript, Piscataway, NJ, USA). On day 0, before the intraperitoneal injection, and on day 2, twenty-four-hour urine samples were collected in metabolic cages (Tecniplast, Buguggiate VA, Italy). The urine samples were collected in tubes containing protease inhibitors (5 × 10^−4^ M PMSF, 10^−6^ M leupeptin) and a bacteriostatic preservative, 3 × 10^−4^ M NaN_3_. At the end of day 3, all animals were overdosed with anesthesia, and then their thoraxes were opened and their blood was drawn via cardiac puncture to preserve the serum of each rat. Figure 2 shows a schematic of the experimental protocol.

### 4.5. Metabolic Characterization

Enzymatic colorimetric assays were utilized to measure total cholesterol (Pointe Scientific, Warszawa, Poland), free cholesterol (DiaSys Diagnostic Systems GmbH, Holzheim, Germany), triacylglycerol (Wiener Lab, Rosario, Argentina), and phospholipid concentrations (Wako Diagnostics, Mountain View, CA, USA). The cholesteryl ester mass content of the lipoprotein fractions was calculated as the difference between total cholesterol and free cholesterol multiplied by 1.67. Creatinine concentrations in serum and urine were measured using the enzymatic method (Pointe Scientific, Warsaw, Poland). The glomerular filtration rate was calculated based on the creatinine clearance. Enzyme-linked immunoassay kits were utilized to measure the levels of rat proteins, including albumin (AssayPro, ERA3201-1) and nephrin (Wuhan EIAab Science Co., Ltd., E0937r Wuhan, Hubei, China), as well as metabolites such as 8-OHdG (Cayman Chemical, kit 589320-96, Ann Arbor, MI, USA) and 8-iso-PGF2α (Wuhan Fine Biotech Co., Ltd., ER1580, Wuhan, Hubei, China). Urinary metabolites and protein excretion were measured in samples collected from rats that were individually housed in metabolic cages for 24 h. Urine volume was determined using gravimetric measurements.

### 4.6. Statistical Analysis

Statistical analyses were conducted using GraphPad Prism (ver. 4.0) software. The Shapiro–Wilk test was utilized to determine the normality of the distribution of the variables. Continuous variables with a normal distribution were reported as means ± standard error of the means. A paired *t*-test was used to assess changes in repeated measures. The statistical significance between the groups was determined using one-way ANOVA and post hoc Tukey’s multiple comparisons. Differences were considered significant at *p* < 0.05.

### 4.7. Materials

All agents were purchased from Avantor Performance Material Poland S.A. (Gliwice, Poland).

## 5. Conclusions

In conclusion, oxidatively modified LDL increases urinary albumin excretion, probably by acting on glomerular filtration cells, and this may lead to a decrease in renal function.

## Figures and Tables

**Figure 1 ijms-25-05498-f001:**
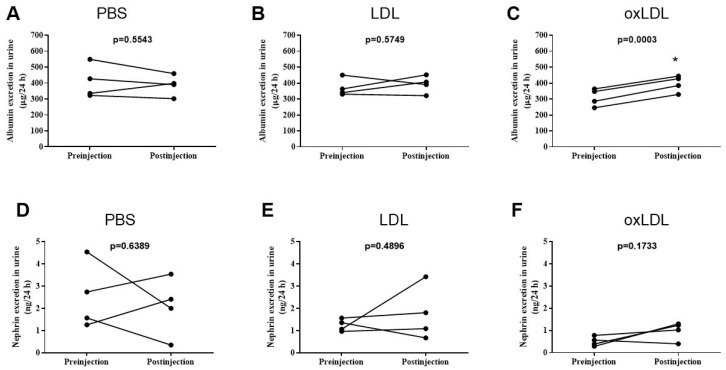
The effects of phosphate-buffered saline (PBS), low-density lipoprotein (LDL), or oxidized low-density lipoprotein (oxLDL) on urinary albumin (**A**–**C**) and nephrin (**D**–**F**) excretion.* *p* < 0.05 vs. preinjection.

**Figure 2 ijms-25-05498-f002:**
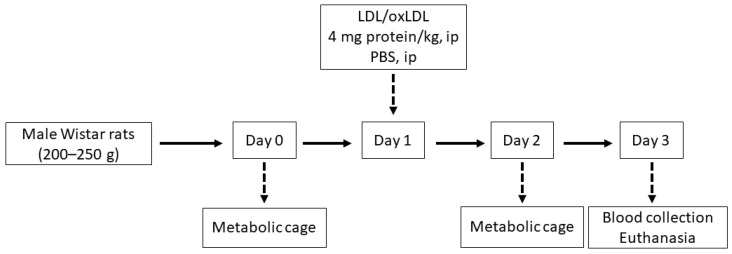
Experimental procedure scheme.

**Table 1 ijms-25-05498-t001:** Clinical and biochemical characteristics of patients on LDL apheresis.

Gender	Female
Lp(a) before first LA, mg/dL	137
Age of first LA, years	55
Year of starting LA	2013
Coronary artery disease	yes
Age of coronary artery disease diagnosis, years	48
Acute coronary syndrome	No
Acute coronary syndrome, age of first	NA
Percutaneous coronary intervention	3
Age of first percutaneous coronary intervention	48
Coronary artery bypass graft	no
Coronary artery bypass graft, age	NA
Transient ischemic attack	no
Stroke	no
Stroke, age of first attack	NA
Carotid artery disease	no
Peripheral artery disease	no
Revascularization of carotid or peripheral artery	no
BMI, body mass index	24
Heterozygous familial hypercholesterolemia	yes
Diabetes	no
Hypertension	yes
Smoking history	yes
Family history of early ASCVD in 1st-degree relative	yes
Chronic kidney disease	no
LVEF, left-ventricle ejection fraction (%)	60

NA-not applicable.

**Table 2 ijms-25-05498-t002:** Composition of low-density lipoproteins isolated from a patient with familial hypercholesterolaemia who underwent LDL apheresis.

Concentration (Relative Amount)	Low-Density Lipoprotein Compounds				
	Phospholipids	Triacylglycerols	Free Cholesterol	Esterified Cholesterol	Proteins
mg/dL	305.00 ± 9.80	104.20 ± 2.60	139.70 ± 7.80	610.90± 23.30	335.70 ± 16.60
(%)	24.40 ± 0.20	7.00 ± 0.20	9.30 ± 0.20	40.90 ± 0.50	22.40 ± 0.40

Results are expressed as means ± standard error of the means (n = 6).

**Table 3 ijms-25-05498-t003:** Characterization of native LDL (nLDL) and oxidized LDL (oxLDL). Low-density lipoproteins isolated from a patient with FH who had undergone LA (native LDL) were oxidized in vitro (oxLDL) in the presence of 5 µM of copper (II) sulfate for 24 h.

Parameter	Lipoprotein	
	nLDL	oxLDL
TBARS (nmol/mg LDL protein)	0.92 ± 0.16	6.36 ± 0.64 *
Conjugated dienes (OD at λ = 234 nm)	0.49 ± 0.02	1.59 ± 0.03 *
Free ε-amino groups (nmol/mg LDL protein)	400.40 ± 13.62	149.10 ± 28.86 *

Abbreviations: TBARS, thiobarbituric-acid-reactive substances; nLDL, native LDL; oxLDL, oxidized LDL. Results are expressed as means ± standard error of the means (n = 4 per group). * *p* < 0.0001 (unpaired *t*-test).

**Table 4 ijms-25-05498-t004:** Effects of LDL or oxidized LDL (oxLDL) on cholesterol, triacylglycerols, and TBARS concentrations and creatinine clearance in rats. Rats were intraperitoneally injected with sterile lipoprotein suspensions at a dose of 4 mg of protein per kg of body weight.

Parameter	Experimental Groups		
	PBS	LDL	oxLDL
Cholesterol (mg/dL)	82.00 ± 12.00	75.00 ± 4.00	71.00 ± 9.00
Triglycerides (mg/dL)	121.00 ± 21.00	84.00 ± 31.00	132.00 ± 44.00
TBARS (µmol/L)	2.32 ± 0.49	2.09 ± 0.36	1.82 ± 0.36
Creatinine clearance (mL/min)	2.15 ± 0.18	2.42 ± 0.49	2.27± 0.47

Abbreviations: TBARS, thiobarbituric-acid-reactive substances; oxLDL, oxidized LDL. Results are expressed as means ± standard error of the means (n = 4 for each group).

**Table 5 ijms-25-05498-t005:** Diuresis and urinary excretion of oxidative stress markers in rats after administration of LDL or oxidized LDL (oxLDL). The rats received intraperitoneal injections of PBS and either LDL or oxLDL at a dose of 4 mg of protein per kg of body weight. Urine samples were collected in metabolic cages 24 h before (pre-injection) and 24 h after (postinjection) the infusion of lipoproteins and PBS.

Urinary Excretion	Experimental Groups					
	PBS		LDL		oxLDL	
	Pre-Injection	Postinjection	Pre-Injection	Postinjection	Pre-Injection	Postinjection
Water (mL/24 h)	8.00 ± 1.00	8.00 ± 2.00	10.00 ± 1.00	12.00 ± 3.00	9.00 ± 2.00	9.00 ± 1.00
TBARS (µmol/24 h)	67.00 ± 11.00	68.00 ± 6.00	97.00 ± 8.00	96.00 ± 17.00	93.00 ± 14.00	93.00 ± 12.00
8-iso-PGF2α (ng/24 h)	3.65 ± 1.89	3.89 ± 0.98	3.39 ± 0.86	2.88 ± 0.67	3.36 ± 1.57	3.00 ± 0.87
8-OHdG (µg/24 h)	1.86 ± 0.13	1.85 ± 0.28	2.19 ± 0.38	2.57 ± 0.83	1.78 ± 0.33	1.82 ± 0.10

Abbreviations: TBARS, thiobarbituric-acid-reactive substances; oxLDL, oxidized LDL; 8-iso-PGF2α; 8-isoprostane; 8-OHdG, 8-hydroxy-2′-deoxyguanosine. Results are expressed as means ± standard error of the means (n = 4 for each group).

## Data Availability

The data presented in this study are available on request from the corresponding author.

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
