# Peer review of "Human In Vitro Oxidized Low-Density Lipoprotein (oxLDL) Increases Urinary Albumin Excretion in Rats"

_ijms, 2024, doi:10.3390/ijms25105498_

Round 1
Reviewer 1 Report
Comments and Suggestions for Authors
The manuscript entitled “Human in vitro oxidized low-density lipoprotein (oxLDL) in- 2 creases urinary albumin excretion in rats” provides interesting information regarding how oxidatively modified LDL increases urinary albumin excretion, and this may disrupt renal function. However, authors can include more information and following are my concerns:
1.) Line 87: please include total number of patients (men and women) included in this study along with the age range of all these patients.
2.) Please include also if authors observed any history for other diseases such as CVD, diabetes, or stroke in patients included in this study.
3.) Did authors look for any other factors such as BMI, fasting plasma glucose, hemoglobin A1c, or any other factor such as cytokines IL-6, IL-8.
4.) Line 104: Injection of PBS and LDL did not have significant effect urinary albumin excretion 104 (Ualb), Please check this line.
5.) Did authors observed any change in weight or any death of rats during this experiment. That would be great if they can include data regarding weight of rats before and after injection.
6.) Did all the participants (Patients from whom sample was collected for this study), provided written informed consent for inclusion in the study ?
7.) Please include how authors compare human patient data with normal, as they mentioned samples collected from patients. Please mention the control/normal data that used to compare patient data.
Comments on the Quality of English Language
Minor editing of English language and grammar is required.
Author Response
Proszę zobaczyć załącznik.

Reviewer 2 Report
Comments and Suggestions for Authors
The article reports on the examination of the impact of exogenous oxLDL on urinary excretion of albumin and nephrin. The authors achieved this by constructing an in vivo experimental animal model of rats were treated with a single intraperitoneal injection of PBS, LDL, or oxLDL. The study design is appropriate and contains a number of examination methods to analyze the effects of oxLDL on urinary albumin excretion.
Observations:
- In Table 1, 2, 3 and 4 it would be advantageous the same amount of decimals (instead of 10 use 10.00).
- In Figure 1. The significance values should be in bold font and the use of “.” instead of “,” is advise. In addition, on Figure 1. Graph C, use markers to show the significance, otherwise the main take home message of the graph is lost.
- Line 153 and 161 the “in vivo” and “in vitro” should be in italics since it’s a terminus technicus.
- In future experiments I would advise to increase the animals involved in the experimental groups, since n=4 is too low of a number to properly analyze the data from a statistical standpoint.
The use of English language is appropriate; the message of the article comes through clean from the text.
Round 2
Reviewer 1 Report
Comments and Suggestions for Authors
This revised version of manuscript is more clear and thank you authors for revising a manuscript for publication. Please include a table related to patients history in manuscript text as well.
Line 68-70: Please rephrase these lines.
Comments on the Quality of English Language
English language is fine for this manuscript.
Author Response
This revised version of manuscript is more clear and thank you authors for revising a manuscript for publication. Please include a table related to patients history in manuscript text as well.
We would like to thank the Reviewer for the positive evaluation of our revised manuscript. We have now included a medical history of the patient into the manuscript
Table 1. Clinical and biochemical characteristics of the patient on LDL apheresis
|
Gender |
Female |
|
Lp(a) before first LA, mg/dl |
137 |
|
Age of first LA, years |
55 |
|
Year of starting LA |
2013 |
|
Coronary artery disease |
yes |
|
Age of coronary artery disease diagnosis, years |
48 |
|
Acute coronary syndrome |
No |
|
Acute coronary syndrome, age of first |
NA |
|
Percutaneus coronary intervention |
3 |
|
Age of first percutaneus coronary intervention |
48 |
|
Coronary artery bypass graft |
no |
|
Coronary artery bypass graft, age |
NA |
|
Transient ischemic attack |
no |
|
Stroke |
no |
|
Stroke, age of first |
NA |
|
Carotid artery disease |
no |
|
Peripheral artery disease |
no |
|
Revascularization of carotid or peripheral artery |
no |
|
BMI, body mass index |
24 |
|
Heterozygous familial hypercholesterolemia |
yes |
|
Diabetes |
no |
|
Hypertension |
yes |
|
Smoking history |
yes |
|
Family history of early atherosclerotic cardiovascular disease in 1st-degree relative |
yes |
|
Chronic kidney disease |
no |
|
LVEF, left ventricle ejection fraction (%) |
60 |
Line 68-70: Please rephrase these lines.
With the aim of improving clarity, we have rephrased the text to read as follows.
“It is possible that a disruption in podocyte number or function may lead to an increase in protein, such as albumin, passing through the glomerular filter into the urine, which could potentially affect the tubular cell function and result in an increased concentration of albumin in the urine”.
Comments on the Quality of English Language
English language is fine for this manuscript.
